# An integrated workflow for the structure elucidation of nanocrystalline powders
Chiara Sabena [1], Federica Bravetti [2], Natsuki Miyauchi[3], Miho Nakafukasako[3], Yoshitaka Aoyama [3], Katsuo Asakura[3], Kiyotaka Konuma[3], Masahiro Hashimoto[3], Yusuke Nishiyama [3] ✉ & Michele R. Chierotti [1] ✉

Structural characterization of powder materials, including those synthesized by mechanochemical methods, remains challenging due to the lack of single crystals suitable for X-ray diffraction. Microcrystal-Electron Diffraction (MicroED) enables structure determination from sub-micrometer crystallites but faces limitations, particularly in locating hydrogen atoms and distinguishing light atoms (C, N, O). We present a general workflow that integrates MicroED with high-resolution mass spectrometry, database mining, solution and solid-state NMR, and DFT-D/GIPAW calculations to resolve atomic structures of complex powders, even with unknown composition. The approach is demonstrated on a pyridoxine-N-acetyl-L-cysteine salt, a mechanochemically synthesized adduct for which large single crystals could not be obtained, and on N-formyl-methionyl-leucyl-phenylalanine (fMLF), a bacterial chemoattractant peptide. This strategy enables comprehensive structure resolution, including identification of molecular components, crystal packing, atom assignments and hydrogen positions. Its modularity and scalability make it suitable for a wide range of powder materials, e.g., pigments, pharmaceutical compounds, etc., especially when conventional crystallography fails.

Mechanochemical syntheses have gained significant attention for their efficiency, sustainability, and minimal solvent requirements, aligning with Green Chemistry principles[1–5]. In fact, mechanochemistry, which involves the use of mechanical energy to induce chemical reactions[1], offers several advantages over traditional solvent-based crystallization methods. It enables the formation of crystalline products that cannot be obtained by traditional methods in a green way, and it often provides quantitative reactions. Specifically, mechanochemical techniques can be commonly applied in "dry", i.e., grinding or ball milling, as well as "wet" conditions, known as solvent-drop grinding (SDG)/liquid-assisted grinding (LAG)/kneading[1,2]. The "dry" methods consist in a direct grinding of chemical reagents manually in a mortar and pestle or using ball mill devices, typically in milligram-scale amounts[6–8]. The "wet" techniques also involve the addition of small amounts of solvent (typically few drops), which can further improve the efficiency of mechanochemical reactions by providing a lubricant for molecular diffusion[2–4,7].

Mechanochemical synthesis has found wide applications in the environmentally friendly preparation of multicomponent molecular crystals, including cocrystals, salts, and solvates. In pharmaceutical science, these adducts have become essential for modulating and optimizing key physicochemical properties, e.g., solubility, stability, dissolution rates, and bioavailability, of Active Pharmaceutical Ingredients (APIs), making them an attractive alternative to traditional drug formulations[9–20].

However, despite the significant advantages of mechanochemistry, it also presents challenges, mainly related to structural characterization. Single-crystal X-Ray diffraction (SCXRD) remains the most widely used method for determining the atomic structure of crystalline materials[2], but it requires high-quality crystals that cannot be obtained using mechanochemical approaches which usually provide nano- to micro-crystalline powders rather than large, well-formed single crystals[16,19,21]. Additionally, mechanochemical methods tend to produce metastable or amorphous phases, further complicating the ability to grow single crystals suitable for diffraction studies[1,2,4]. To overcome these issues, powder X-Ray diffraction (PXRD) has been widely employed for the structural characterization of pharmaceutical adducts. Recent advancements in PXRD techniques and data analysis, *via* direct-space or more sophisticated methods, have enhanced its capability to solve complex structures, even when single crystals are unavailable[22–25]. In addition, solid-state NMR (SSNMR), within the NMR Crystallography discipline, has emerged as a powerful approach for elucidating the structures of pharmaceutical solids, either alone or in combination with PXRD data and/or computational methods (DFT-D and crystal structure prediction) to derive detailed structural information[24,26–30].

[1]Department of Chemistry, University of Turin, Turin, Italy. [2]Institute of Inorganic and Analytical Chemistry, Goethe University, Frankfurt am Main, Germany. [3]JEOL Ltd., Akishima, Tokyo, Japan. ✉e-mail: yunishiy@jeol.co.jp; michele.chierotti@unito.it

Beyond crystalline solids, SSNMR has also proven invaluable for characterizing partially ordered or amorphous systems where crystallization is challenging, such as polymers, gels, or soft materials, due to its sensitivity to local structure and hydrogen bonding[31–37].

In recent years, microcrystal-electron diffraction (MicroED) has emerged as alternative technique capable of overcoming the limitations imposed by the absence of large single crystals[38–41]. It enables the determination of atomic-resolution structures from crystals as small as a few hundred nanometers in size, i.e., several orders of magnitude smaller than those needed for SCXRD[39,42–46], thanks to its stronger interaction with matter and reduced damage per scattering event compared to X-rays[47–49]. This capability is particularly advantageous for crystalline materials produced through mechanochemical synthesis, which is why it has found widespread application in both materials science and pharmaceutical research, offering several key benefits[42,44,45,50–53]. However, some notable challenges include the imprecise location of hydrogen atoms, due to their low scattering power, and the ambiguous assignment of carbon, nitrogen, and oxygen atoms, due to their similar atomic numbers[39]. In addition, MicroED analysis is performed on only a few single crystals selected from a bulk powder sample composed of numerous microcrystals. Therefore, the determined crystal structure may not necessarily represent the dominant phase of the entire sample since there remains a possibility that the structure corresponds to a minor crystalline component within the mixture.

To address these issues, it was demonstrated that SSNMR can serve as a powerful complementary method often combined with the first-principle quantum computation in the NMR Crystallography framework[38]. Its sensitivity to subtle changes in the electronic environment, particularly those involving hydrogen atoms, enables precise determination of hydrogen positions along hydrogen bonds, making SSNMR highly complementary to diffraction techniques. This capability is especially important for distinguishing between salt and cocrystal forms[54–56], as well as for identifying the tautomeric[24,57] and zwitterionic character[28]. Such detailed structural information is crucial for the rational design and performance optimization of pharmaceutical crystal forms as well as for intellectual property and patent purposes[58–65]. Additionally, SSNMR observations can confirm if the MicroED-derived structure is representative of the entire sample, making these techniques extremely complementary.

To facilitate this analysis, high-resolution mass spectrometry (HRMS) can provide complementary information. However, to the best of our knowledge, the combined use of HRMS and MicroED has not yet been demonstrated. HRMS enables precise determination of the molecular formula, including hydrogen atoms, through sub-digit mass accuracy, offering critical insights for MicroED-based structural analysis. Additionally, HRMS can confirm the formation of supramolecular and molecular complexes, and helps to assign carbon, nitrogen and oxygen atoms in MicroED-based structure.

In this article, we propose a systematic approach for the structure elucidation of nano-crystalline samples by integrating MicroED, HRMS, database mining, solution/SSNMR, and calculations. First, the use of 1D $^{13}C$ SSNMR enables the determination of the number of molecules in the asymmetric unit cell, which facilitates the interpretation of MicroED data, despite inherent ambiguities in distinguishing carbon, nitrogen and oxygen

atoms. HRMS provides the molecular formula and allows assessment of supramolecular adduct formation. The molecular formula can then be used to query chemical structure databases (such as PubChem, ChemSpider, ChEMBL, DrugBank, and Reaxys) to generate a set of candidate structures. By comparing these candidates with the molecular skeleton obtained by MicroED, the list can be efficiently narrowed down to a few likely candidates. The remaining ambiguity can easily be resolved using 1D solution NMR. This information helps to speed up the MicroED structure solution. Finally, SSNMR allows evaluating and refining the crystalline structure including precise hydrogen positions through an NMR Crystallography approach. The method is first demonstrated for the structural determination of a pyridoxine (PN) and N-acetyl-L-cysteine (NAC) adduct (Fig. 1), a previously reported nutraceutical-API salt synthesized via mechanochemical methods[66].

PN-NAC can be synthesized only through the dry grinding technique, and countless attempts to obtain suitable single crystals, even using various crystallization solvents, were unsuccessful, due to the sticky nature of the product (experimental details in Methods section and in the Supplementary Methods). This offers a perfect case of study in which the integration of MicroED with spectroscopic and computational techniques proves to be a powerful and complementary approach for comprehensive structural characterization of crystalline adducts. Moreover, to further illustrate that the method is generalizable, the same procedure was applied for the structure elucidation of N-formyl-methionyl-leucyl-phenylalanine (fMLF, Fig. 1), a bacterial chemoattractant peptide[67,68], whose structure has not yet been reported in literature, despite its widespread use as a model compound in solid-state NMR methodology owing to its well resolved $^{13}C$ and $^{1}H$ peaks. This approach enables the development of a comprehensive workflow, akin to a hypothetical 'blind test', starting from an unknown sample and culminating in the elucidation of its solid-state structure.

## Results and discussion
### Pyridoxine–N-acetyl-L-cysteine salt
The step-by-step analysis of PN–NAC provided the opportunity to present the comprehensive workflow of this approach, illustrated in the flowchart in Fig. 2. In particular, this generalizable strategy enables the refinement of structures obtained by MicroED, effectively addressing two major limitations: the invisibility of hydrogen atoms and the challenge of distinguishing between carbon, nitrogen, and oxygen atoms. It integrates high-resolution MS, chemical structure database analysis, solution NMR, DFT-D and GIPAW calculations, SSNMR, and powder X-ray diffraction (PXRD). To enhance the generalizability of the approach, the sample is initially treated as an unknown compound ("compound **1**"), without any prior assumption about its composition. All aspects, e.g., its composition, number of components, and structural features, are gradually resolved through the integrated analytical approach.

Initial structural information of compound **1** was derived from MicroED analysis of the powder sample (Fig. 3), yielding the preliminary structure shown in Fig. 4a and summarized in Supplementary Table S5. The structure clearly reveals the presence of two distinct molecular species, A and B, forming a binary molecular adduct (A + B). In addition, two sets of

**Fig. 1 | Chemical structures of the studied compounds.** Chemical structures of pyridoxine (PN, left), N-acetyl-L-cysteine (NAC, middle) and N-formyl-methionyl-leucyl-phenylalanine (fMLF, right), with atom numeration.

pyridoxine
**(PN)**

N-acetyl-L-cysteine
**(NAC)**

N-formyl-methionyl-leucyl-phenylalanine
**(fMLF)**

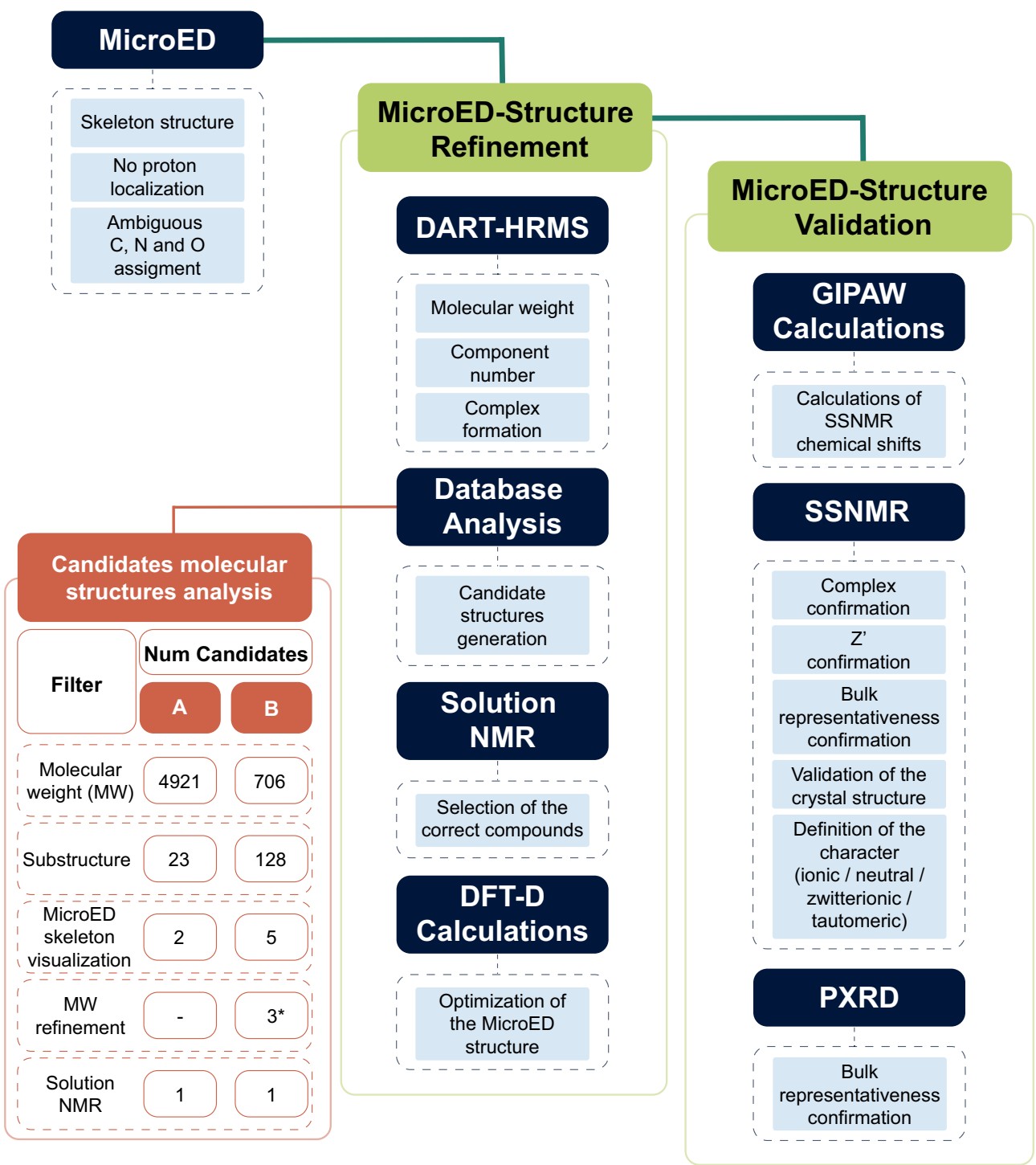

**Fig. 2 | Flowchart illustrating the proposed approach.** Schematic representation of the workflow used in this study. The database analysis was performed using the PubChem library. A and B are two generic compounds whose presence is confirmed by DART-HRMS: in our case, numbers refer to the present study (i.e., to the structure solution of PN and NAC). The asterisk highlights that all the three structures obtained for compound B (i.e., NAC), after molecular weight refinement filter, are isostructural crystals differing in absolute structure.

this unit formula are found in the asymmetric unit (i.e., Z' = 2). As expected, hydrogen atoms were not observed in the electrostatic potential map, and permutations among carbon, nitrogen, and oxygen atoms led to only marginal improvements in the R1 factor. Therefore, no tentative assignments were made at this stage and non-hydrogen atoms were initially represented in gray, while a heavier atom, sulfur, was also identified. While a more advanced MicroED data analysis including dynamic scattering[69] might partially address these issues, we propose here an alternative and

straightforward strategy for resolving hydrogen positions and elemental assignments of carbon, nitrogen and oxygen atoms.

To facilitate structure elucidation, HRMS was performed. Among various ionization techniques, we employed direct analysis in real time (DART)-HRMS, which enables rapid and soft ionization under ambient conditions. DART-HRMS allows for the direct measurement of solid powder samples simply by placing them into the ion source, eliminating the need for prior sample preparation. This feature is particularly advantageous

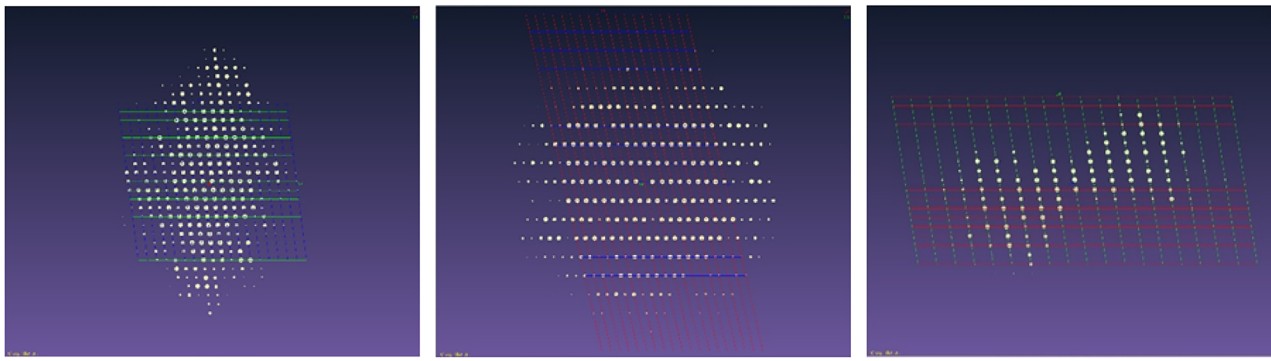

**Fig. 3 | Reconstructed 3D reciprocal lattice of compound 1.** Reconstructed three-dimensional reciprocal lattice of compound 1 obtained from MicroED data, shown along the a* (left), b* (middle), and c* (right) axes.

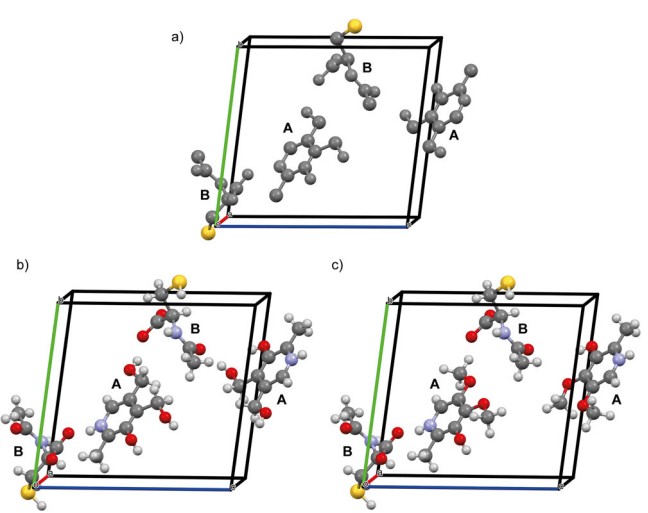

**Fig. 4 | Asymmetric unit of compound 1 (PN-NAC). a** Preliminary MicroED-derived structure of compound 1, revealing two distinct molecular species (A and B) in a 1:1 stoichiometric ratio and two independent unit formula (Z′ = 2). Carbon, nitrogen, and oxygen atoms are undifferentiated (gray), hydrogen atoms are not visible, and a heavier atom consistent with sulfur (yellow) is identified. **b** Refined structure after integration of HRMS, database filtering, and NMR analysis. Molecular species A and B are assigned to pyridoxine (PN) and N-acetyl-L-cysteine (NAC), respectively, with all non-hydrogen atoms correctly labeled and colored. **c** Rejected candidate structure for molecular species A, excluded based on DEPT-135 solution NMR data, that confirmed the presence of two -CH₂ groups instead of methoxy moieties.

when analyzing insoluble or unstable compounds. Importantly, the absence of sample handling prevents any alternation or degradation during sample preparation. The DART-HRMS spectra of the dry powder of compound **1** (Supplementary Fig. S1) revealed two different ion species, $[A + H]^+$ and $[B - H]^-$, in full agreement with the MicroED structure. Additionally, adduct species, such as $[A + B + H]^+$ and $[A + B - H]^-$, were also detected, further supporting the multicomponent crystal formation. Monoisotopic peaks were observed at 170.0815 Da for $[A + H]^+$ and 162.0231 Da for $[B - H]^-$, suggesting $[A] = 169.0742$ Da and $[B] = 163.0304$ Da. This is consistent with molecular formulas of $C_8H_{11}NO_3$ (169.0739 Da) for A and $C_5H_9NO_3S$ (163.0303 Da) for B. These assignments aligned with the number of non-hydrogen atoms and the presence of sulfur observed in the MicroED data (Fig. 4a).

To further refine the identity of A and B, we made a query into the PubChem database. Firstly, the average molecular weights, considering the natural abundance of each isotope, were calculated: based on the molecular formulas, 169.1778 Da for A and 163.1949 Da for B were obtained. Initial searches, based solely on molecular formulas, returned an unmanageable number of hits (4921 for A and 706 for B), hindering the refinement of the

MicroED structure. Therefore, we applied additional filters: the number of non-hydrogen atoms (12 for A and 10 for B based on MicroED data, Fig. 4a) and relevant substructural features. Specifically, the aromatic ring in A was modeled as either a pyridine (A-Pyr) or benzene ring (A-Ben), while the sulfur atom in B was assumed to be part of either a C-S (B-C), O-S (B-O) or N-S (B-N) bond, as the MicroED structure revealed the presence of X-S (X = C,N,O) bonding in B. The queries used are listed in the Supplementary Notes (Paragraph 2.2): a ± 0.006 Da margin was included, as PubChem reports molecular weights with a precision of 0.01 Da. These criteria drastically reduced the number of candidate structures: 18 for A-Pyr, 5 for A-Ben (i.e., a total of 23 for A), 33 for B-C, 78 for B-O, and 17 for B-N (i.e., a total of 128 for B). All possible isomers are reported in Supplementary Fig. S2. This demonstrates how integrating HRMS data into MicroED analysis, guided by a chemical structure database, can significantly narrow the search space for structure refinement. Finally, we performed a visual comparison between the MicroED-derived molecular skeleton and the remaining candidate structures to exclude those that were unfeasible, resulting in 2 and 5 candidates for A and B, respectively (Supplementary Fig. S3).

Among the 5 candidates for B, two were excluded due to discrepancies in the molecular formula ($C_4H_9N_3O_2S$, 163.0415 Da), which is 0.011 Da heavier molecular weight than the HRMS one ($C_5H_9NO_3S$). Since the difference is much larger than TOF MS precision, we could safely exclude these two possibilities. This inconsistency arises from the limited precision of the molecular-weight data and the query constraints available in PubChem, e.g., it is not possible to simultaneous constraints on molecular formula and substructures. Databases offering higher numerical resolution and the possibility to combine molecular formula and structural substructure filters (such as Reaxys or CAS SciFinder) could overcome this limitation and yield more accurate and specific results, although access requires a paid license. The remaining three B candidates were stereoisomers, differing only for the absolute structure, therefore, based on the natural abundance, we assigned B as N-acetyl-L-cysteine (NAC). At this point, all the carbon, nitrogen, and oxygen atoms could be unambiguously assigned (Fig. 4b, c). Moreover, we could also locate the hydrogen atoms using riding models. It is worth noting that absolute structures cannot be determined by the current MicroED analysis as we used kinetic analysis. However, we cannot exclude the possibility of N-acetyl-D-cysteine. Further analyses, such as dynamic scattering analysis together with fine step MicroED data collection, would be required to address this issue, as these approaches can directly provide information on the absolute configuration without the need for additional analytical methods[70]. This is out of scope of the current paper and no further analysis was conducted here.

Two possible candidates remained for A. To discriminate between them, we employed solution NMR spectroscopy. Specifically, 1D $^{13}$C-NMR was used to identify the presence of functional groups expected in each candidate. For both proposed, we expected two peaks in the 50–60 ppm range, associated with -CH₂-OH or -O-CH₃ moieties (Fig. 4b, c, respectively). A DEPT-135 experiment was performed to distinguish CH₂ and CH₃ carbons. As shown in Supplementary Fig. S6, the two signals (at 58.2

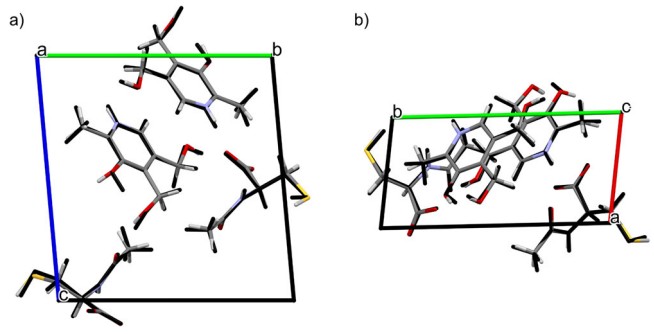

**Fig. 5 | Overlay of the experimental and DFT-D optimized crystal structures of PN-NAC.** Overlay of the experimental crystal structure of PN-NAC (shown in standard atom colors) with the DFT-D optimized structure (shown in black), viewed along the **a** a-axis and **b** c-axis.

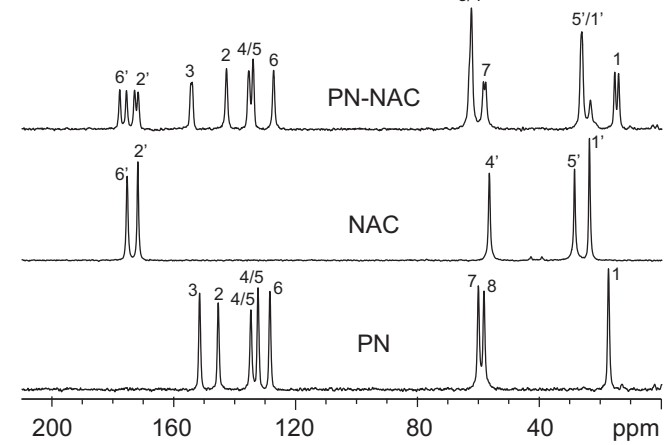

**Fig. 6 | $^{13}$C CPMAS SSNMR spectra of PN-NAC, NAC and PN.** $^{13}$C (150.9 MHz) CPMAS SSNMR spectra of the PN-NAC salt (top), pure N-acetyl-L-cysteine (NAC, middle) and pure pyridoxine (PN, bottom), acquired at room temperature at a spinning speed of 20 kHz. Atom numeration refers to Fig. 1.

and 56.8 ppm) were identified as $CH_2$ groups (negative phase), ruling out the presence of $-O-CH_3$ groups and thus excluding the structure shown in Fig. 4c. Accordingly, molecule A was confidently assigned to pyridoxine (PN) (Fig. 4b). The remaining CH signal (at 56.5 ppm) corresponds to the Cα carbon of NAC. While the DEPT-135 experiment was sufficient for this particular case, the required information largely depends on the system and it might require $^1$H-NMR, $^{13}$C-NMR or even 2D correlation NMR techniques. It is important to emphasize that the current workflow presupposes that the compounds present in the crystal are included in existing chemical structure databases. The potential presence of previously unreported species cannot be entirely excluded. Therefore, a full validation of the structural assignments *via* solution NMR remains essential. For this purpose, we acquired a comprehensive set of solution NMR spectra, including $^1$H-NMR, $^{13}$C-NMR, $^1$H{$^{13}$C} HMQC, $^1$H{$^{13}$C} HMBC, and $^{13}$C DEPT-135 (Supplementary Figs. S4–S7). All the peaks were unambiguously assigned (Supplementary Table S4), providing strong confirmation of the proposed molecular identities. In cases where database-based searches yield ambiguous or inconsistent results, solution NMR becomes the primary tool for structural elucidation.

To further improve the accuracy of atomic coordinates and hydrogen positions, the MicroED-derived structure was geometry-optimized by using DFT-D calculations, optimizing both atomic positions and lattice parameters. The optimized unit cell parameters (reported in Supplementary Table S5) show a good agreement with the experimental ones. We observed a reduction of 4.7% of the unit cell volume of the optimized crystal structure, which is within the expected values (4–5%) for DFT-D calculations performed at 0 K[71]. The optimized structure (provided as Supplementary Data 1 and 2, optimized with variable and fixed lattice parameters, respectively) and its comparison with the previous experimental one is shown in Fig. 5. No significant differences in the molecular conformation nor in the local molecular arrangement were observed. The calculated RMSDC (Root-Mean Square Deviation Cartesian) value is as low as 0.175 Å, further indicating that the experimental structure is correct[72].

Subsequently, we also calculated chemical shifts by GIPAW calculations performed on the DFT-D optimized structure with fixed lattice parameters and compared them with SSNMR experimental $^{13}$C and $^1$H chemical shifts (Supplementary Table S6 and Supplementary Figs. S8–S10). As for the experimental $^{13}$C CPMAS SSNMR spectrum (Fig. 6), it exhibited well-resolved signals, consistent with a 1:1 stoichiometric ratio and confirming both the presence of two crystallographically independent molecules of both PN and NAC in the unit cell and the representativeness of the bulk (as also assessed by PXRD diffractograms, Supplementary Figs. S17, S18). The correspondence between calculated and experimental $^{13}$C chemical shifts was excellent, with a low $^{13}$C root mean square error (RMSE) of 1.9 ppm, further supporting the validity of the proposed structure[26].

Usually, the consistency between calculated and experimental chemical shifts extends to even labile or hydrogen-bonded functional groups (i.e.,

NH groups), except where thermal effects induce discrepancies (i.e., OH groups). Indeed, DFT-D calculations are performed at 0 K, inherently neglecting all the thermal fluctuations. While atomic coordinates generally remain consistent with those at room temperature, OH protons involved in hydrogen bonding often experience temperature-dependent hydrogen positions. In our case, a major difference was observed while comparing the calculated and experimental chemical shifts of H9, H10 and H11, with respect to the others hydrogen atoms as shown in Supplementary Table S6. As a result, the calculated $^1$H RMSE value is 0.8 ppm. This value is higher than expected for a correct crystal structure, usually around 0.3–0.5 ppm for organic molecules, but not completely surprising due to the presence of a considerable number of hydrogen-bonded functional groups in the PN-NAC salt[26]. We also calculated the $^1$H RMSE values excluding H9, H10 and H11 (Supplementary Table S7 and Supplementary Fig. S11). In this case, the $^1$H RMSE value drops to 0.6 ppm. This phenomenon is particularly relevant in pharmaceutical applications, where the exact position of a single hydrogen atom can distinguish between a salt and a cocrystal, with major implications for physicochemical properties and regulatory classification. In the current system, hydrogen H8' is the critical indicator of salt vs cocrystal formation. Indeed, it forms a charge-assisted hydrogen bond with N12 of PN as confirmed both from $^{15}$N CPMAS spectra (Fig. 7a and Supplementary Fig. S15) and the 2D $^1$H/{$^{14}$N} T-HMQC experiment (Fig. 7b). In the formers, a large shift to lower frequencies (Δδ ~86 ppm; from −88.5 to −174.7 ppm) of the signal assigned to N12 is strongly indicative of the protonation from the carboxylic group of NAC and formation of a salt. In the latter, a strong correlation between N12 and H8' is present which allows also to assign the H8' to the signal at 16.4 ppm, further confirmed by additional SSNMR experiments including $^1$H MAS echo, 2D $^1$H{$^{13}$C} short- and long-range DCP and $^1$H DQ/$^1$H SQ (Supplementary Figs. S12–S14). Because the precise location of this hydrogen is critical[54,73], we performed the $^1$H-$^{14}$N PM-S-RESPDOR experiment[60] and the resulting RESPDOR fraction curve achieved on the signal at 16.4 ppm (H8') is shown in Fig. 8.

The analytical fitting gives minimum RMSD at a dipolar coupling of 5.57 kHz, corresponding to a $^1$H–$^{14}$N distance, measured between H8' and N12, of 1.16 Å. This value is perfectly consistent with a covalent N–H bond typically found in protonated nitrogen sites, confirming the salt formation. Indeed, the reference mean values of N-H and N-D distances for pyridine-carboxylic acid N···H···O interactions, extrapolated from a CSD survey (CSD version 5.45, updated in May 2025, Supplementary Fig. S16 in the SI), are 1.11 Å and 1.48 Å for charge assisted and neutral hydrogen bonds, respectively. To further confirm the crystal structure of PN-NAC obtained by MicroED, a Rietveld refinement was performed. All details about refined parameters are given in the Supplementary Methods, while the Rietveld plot

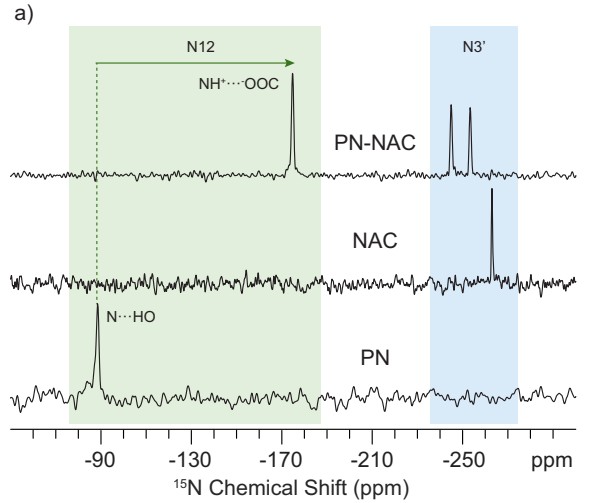

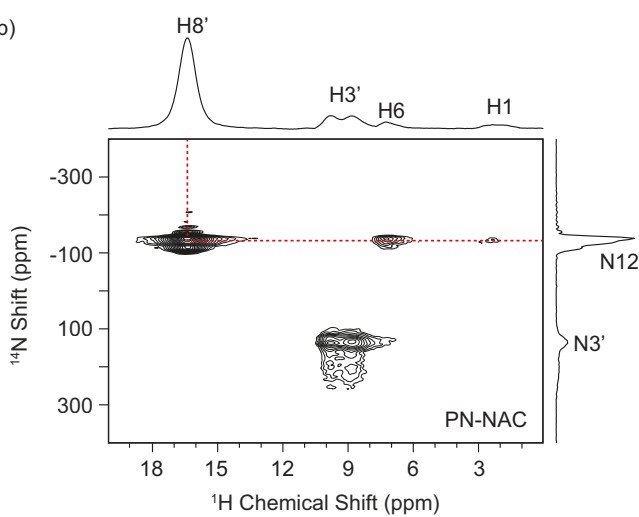

**Fig. 7 | $^{15}$N CPMAS and 2D $^1$H/{$^{14}$N} T-HMQC SSNMR spectra of PN-NAC. a** $^{15}$N (60.81 MHz) CPMAS spectrum of PN-NAC (top), PN (bottom), and NAC (middle), acquired at room temperature at a spinning speed of 15 kHz (PN-NAC and PN) and 12 kHz (NAC). Colored boxes indicate the N12 (green) and N3' (light blue) sites. The green arrow highlights the shift of N12 upon the adduct (salt) formation. **b** 2D

$^1$H/{$^{14}$N} ($^1$H, 600.17 MHz; $^{14}$N, 43.4 MHz) T-HMQC SSNMR spectrum of PN-NAC, acquired at room temperature at a spinning speed of 70 kHz. The red dashed line highlights the correlation between H8' and N12. The $^{15}$N and $^{14}$N chemical shifts are referenced to NO$_2$CH$_3$ and atom numeration refers to Fig. 1.

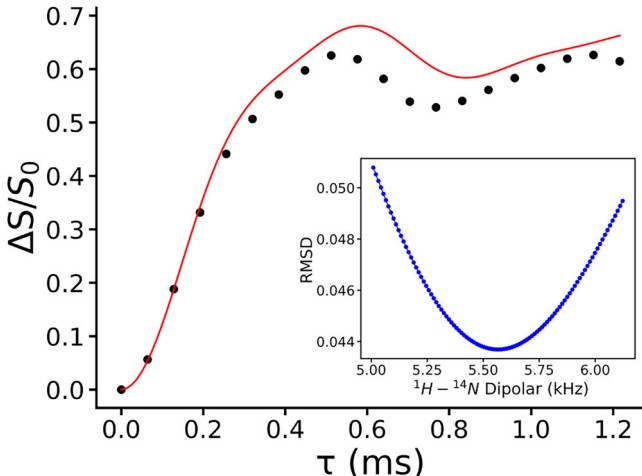

**Fig. 8 | $^1$H-$^{14}$N PM-S-RESPDOR curve of PN-NAC.** The experimental $^1$H$-^{14}$N $\Delta S/S_0$ fraction curve (black dots) achieved by PM-S-RESPDOR of PN-NAC for the $^1$H signal at 16.4 ppm and the best analytical fitting curve (red solid line). The inset shows the $^1$H $- ^{14}$N dipolar coupling fittings based on the root-mean-square deviation (RMSD) analysis.

is reported in Supplementary Fig. S19. The Rietveld refinement confirms that the determined crystal structure for PN-NAC is correct.

## N-formyl-methionyl-leucyl-phenylalanine

To further demonstrate the applicability of the approach, fMLF was taken as a second 'blind test'. The same procedure applied to PN–NAC was followed. The initial structural information on the sample, treated as an unknown compound ("compound **2**"), was obtained from MicroED analysis. This provided a preliminary framework of its structure (Supplementary Fig. S20a), in which hydrogen atoms were not observed, non-hydrogen atoms were initially represented in gray, and a heavier atom, consistent with sulfur, was identified. The application of DART-HRMS (Supplementary Fig. S21) enabled the determination of the molecular formula ($C_{21}H_{31}N_3O_5S$, 437.555 Da), through the observation of the monoisotopic peak at 438.2062 Da, in full agreement with the number of non-hydrogen atoms and the presence of a sulfur atom observed in the MicroED data

(Supplementary Fig. S20a). At this point, database mining was used to assess the identity of compound **2**. Since the initial search, based solely on molecular formula, returned 1399 candidate molecules, further filters were applied: in particular, the number of non-hydrogen atoms (30) and relevant substructural features (i.e., the sulfur atom was assumed to be part of either a C-S-C, O-S-C, N-S-C, O-S-N, N-S-N, or O-S-O bond, as the MicroED structure revealed the presence of X-S-X (X = C,N,O)). This reduced the number of candidates to 75: 14 for C-S-C, 10 for O-S-C, 42 for N-S-C, 9 for O-S-N, and 0 for N-S-N and O-S-O. The corresponding queries are listed in the Supplementary Notes (Paragraph 3.3; a ± 0.05 Da margin was included) and all the possible isomers are reported in Supplementary Fig. S22. A visual comparison between the MicroED molecular skeleton and the remaining candidates allowed to exclude most of them, resulting in three possible candidates: N-formyl-DL-methionyl-DL-leucyl-DL-phenylalanine, N-Formyl-L-methionyl-L-leucyl-L-phenylalanine, and N-formyl-D-methionyl-L-leucyl-L-phenylalanine, which are stereoisomers, differing only in the absolute structure (Supplementary Fig. S23). Both N-formyl-DL-methionyl-DL-leucyl-DL-phenylalanine and N-formyl-D-methionyl-L-leucyl-L-phenylalanine can be excluded, because neither a racemic mixture nor a stereochemical arrangement involving different configurations at the three chiral centers is compatible with the relative molecular geometry observed in the MicroED skeleton. If such species were present, multiple chiral environments or configurational disorder would be evident in the structure, which is not observed. As in the previous PN–NAC system, MicroED data were obtained by using kinetic analysis, which does not allow determination of the absolute configuration and therefore does not allow exclusion of the D-D-D stereoisomer. However, the L-L-L stereoisomer is supported by its biological compatibility, leading to the identification of compound **2** as N-Formyl-L-methionyl-L-leucyl-L-phenylalanine. More advanced MicroED analyses, such as dynamic scattering analysis together with fine step data collection, could provide additional information on the absolute structure[70]. The MicroED-derived structure, with atom assignments refined by the combination of DART-HRMS and the database mining, was then geometry-optimized by using DFT-D calculations with variable lattice parameters. The good agreement between the optimized structure (provided as Supplementary Data 3), and unit cell parameters, with the experimental ones (Fig. 9 and Supplementary Table S8) and the calculated low RMSDC value (0.198 Å) indicate that the experimental structure is correct. GIPAW calculations were subsequently performed on the DFT-D-optimized structure. Computed $^1$H, $^{13}$C, and $^{15}$N SSNMR chemical shifts

**Fig. 9 | Overlay of the experimental and DFT-D optimized crystal structures of fMLF.** Overlay of the experimental crystal structure of fMLF (shown in standard atom colors) with the DFT-D optimized structure (shown in black), viewed along the **a** a-axis and **b** b-axis.

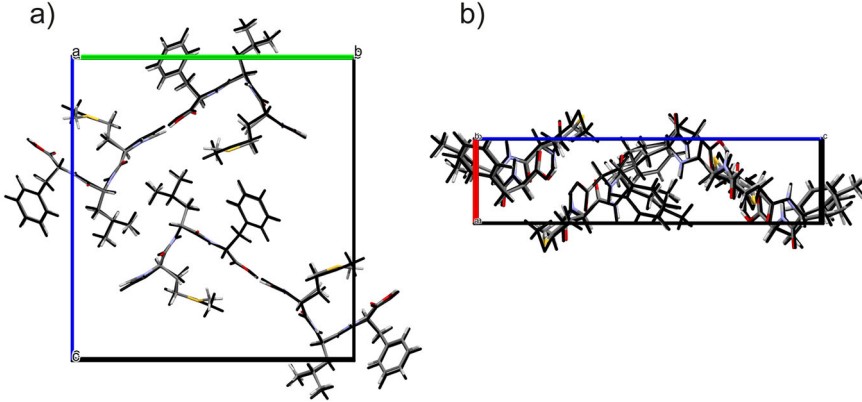

were compared with experimental ones (assigned through several SSNMR experiments including $^1$H MAS echo, 2D $^1$H{$^{13}$C} short- and long-range DCP, $^1$H DQ/$^1$H SQ, and $^1$H{$^{15}$N} DCP; Supplementary Fig. S24–S27), yielding $^1$H, $^{13}$C, and $^{15}$N RMSE values of 0.6, 3.0 and 3.6 ppm (Supplementary Fig. S28–S30), respectively. The slightly higher $^1$H and $^{13}$C RMSE values can be partly attributed to the dynamic behavior of the phenylalanine aromatic ring in fMLF which cannot be considered in the calculations performed at 0 K. Under fast MAS, the phenyl ring undergoes rapid motions, which interfere with the $^1$H-$^{13}$C decoupling and therefore reduce the CP efficiency for aromatic CH carbons[74,75]. As a result, the signals in the 125–135 ppm region are markedly attenuated or even unobservable, making their experimental assignment, and related $^1$H assignment by $^1$H{$^{13}$C} DCP, less robust and leading to a slightly increased RMSE. Importantly, the combination of SSNMR and MicroED data enables to identify this motion as a 180° ring flip. In fact, the phenyl ring is clearly resolved in the ED potential map and would otherwise appear obscured if a continuous rotational motion were present. This observation is fully consistent with previous $^2$H NMR studies on phenylalaine ring dynamics[74], but here the flip is detected without the need for deuterium labeling, highlighting the strength of combining 3D ED with SSNMR. To further verify the effect, RMSE values were recalculated excluding the atoms from the ring (H5/H9 and H6/H8; C4/C5/C6/C8/C9), yielding 0.5 ppm for $^1$H and 2.3 ppm for $^{13}$C (Supplementary Table S10, Supplementary Figs. S31, S32). These low RMSE values confirm the reliability of the obtained structure despite the dynamical attenuation of the aromatic signals. To further confirm the goodness of the crystal structure of fMLF, Rietveld refinement was also performed. The resulting Rietveld plot is reported in Supplementary Fig. S33.

This second example clearly demonstrates the modularity and generalizability of the approach. The general workflow remains the same across different samples; however, the number and depth of analyses performed within each characterization step can be adjusted according to the complexity of the investigated system, in order to extract all the information required for the complete structure elucidation.

## Conclusions

This study demonstrates a generalizable and integrative workflow for the structural elucidation of nanopowder materials. By combining MicroED with HRMS, structure database mining, solution/SSNMR, DFT-D and GIPAW calculations, we address two major limitations of MicroED: the inability to locate hydrogen atoms and the difficulty in distinguishing between carbon, nitrogen, and oxygen atoms in electrostatic maps. The PN–NAC adduct, used here as a case of study, was specifically selected due to its intrinsic complexity: it is a multicomponent system with Z' = 2, synthesized mechanochemically and inaccessible by conventional crystallization methods. These characteristics make it an ideal benchmark for testing the robustness and versatility of the proposed workflow. Through a step-by-step procedure, the system was fully characterized: i) the complete molecular composition and 3D crystal structure were determined; ii) atom identities and hydrogen positions were unambiguously assigned; iii) the salt nature of the adduct was confirmed; iv)

the MicroED-derived structure was validated and shown to represent the bulk material. Furthermore, the comprehensive structure elucidation of fMLF, used as a second example of application of the workflow, enabled to evaluate the generalizability of the method.

It is also worth noting that while some steps of the workflow, such as structure database querying and candidate comparison, were performed manually, they can be easily automated or enhanced through machine-learning tools. Additionally, an intrinsic limitation of database-based approaches is that previously unreported molecules may simply be absent from the database, leading to incomplete or inconclusive searches. In such cases, the workflow could be extended with machine-learning-assisted structure generation, where algorithms propose and rank molecular candidates that satisfy the experimental constraints derived from MicroED, HRMS, and NMR analyses. These predicted structures could then be cross-validated against chemical structure databases such as PubChem, Reaxys, ChemSpider, or CAS SciFinder, thereby broadening the applicability of the method to unknown or novel compounds. Nonetheless, the modular nature of the method makes it adaptable to a broad range of nano- to micro-sized samples, including those of higher complexity, varying stoichiometries, or multiple independent molecules in the unit cell. Overall, this work illustrates the potential of integrating MicroED with complementary spectroscopic and computational techniques to overcome common challenges in the structural analysis of solid forms, paving the way for its broader application in pharmaceutical and materials science, particularly for unknown systems where single-crystal growth is unfeasible.

## Methods
### Adduct synthesis

Pyridoxine-N-acetyl-L-cysteine salt (PN-NAC) was synthetized mechanochemically as previously reported: 100 mg (0.59 mmol) of PN and 96.5 mg (0.59 mmol) of NAC were manually ground for 30 min, and the resulting sticky mixture was placed in a desiccator for one week and then reground to obtain a homogeneous dry powder[66]. N-formyl-methionyl-leucyl-phenylalanine (fMLF) in powder form was purchased from Sigma-Aldrich and used without further purification.

### Characterization techniques

**MicroED**. The ED patterns of the PN-NAC and fMLF crystals were collected using an XtaLAB Synergy-ED (Rigaku corporation and JEOL Ltd., Japan) operating at 200 kV with continuous rotation of the sample. The diffraction data were recorded using a high-sensitivity pixel array detector (Hypix-ED, Rigaku corporation, Japan). The camera length (606.590 mm) was calibrated using a gold polycrystal specimen as a standard.

**DART-HRMS**. DART-HRMS measurements were conducted using JMS-TQ4000GC (JEOL Ltd., Japan) using the ion source of DART (Ion Sense®). $m/z$ reference were calibrated using PEG 600 + 1000 for DART$^+$ and PFPE for DART$^-$.

**Solution NMR.** $^1$H-NMR, $^{13}$C-NMR, $^{13}$C DEPT135, $^1$H{$^{13}$C} HSQC and $^1$H{$^{13}$C} HMBC (with deuterated water, $D_2O$) spectra were acquired on a JEOL ECZR 600 instrument operating at 600.1 MHz.

**DFT-D and GIPAW calculations.** The crystal structure of the PN-NAC salt was optimized at DFT-D level with Quantum Espresso (QE, v. 6.4.1)[76], employing the projector augmented wave (PAW) approach, with the non-local vdW-df2 method[77] and the B86r functional[78] with the SSSP set of pseudopotentials[79]. An energy cut-off of 60 Ry was used. Starting from the optimized structures with fixed lattice parameters, NMR calculations were performed using the Gauge Including Projected Augmented Wave (GIPAW)[80] and the PBE pseudopotentials from PS Library 1.0.0[81] with an energy cut-off of 80 Ry, following the methodology previously described[82,83]. The crystal structure of fMLF was obtained by geometry optimization at the DFT-D2 level using QE (v7.5), employing the PAW approach with verified pseudopotentials from the official QE website. An energy cutoff of 47 Ry, as recommended for the pseudopotentials, was applied. NMR calculations were performed using the GIPAW method with the same cutoff energy of 47 Ry.

**Solid-state NMR.** The $^{13}$C and $^{15}$N CPMAS and $^1$H MAS spectra and 2D $^1$H/{$^{14}$N} T-HMQC, $^1$H{$^{13}$C} double CP (DCP), $^1$H DQ/$^1$H SQ and PM-S-RESPDOR experiments were acquired on a JEOL ECZR 600 instrument, operating at 600.1, 150.9, 60.8, and 43.4 MHz for $^1$H, $^{13}$C, $^{15}$N and $^{14}$N nuclei, respectively. The $^{13}$C chemical shift scales were calibrated through the signals of γ-glycine ($^{13}$C methylenic peak at 43.7 ppm) as an external standard. The $^{15}$N and $^{14}$N chemical shift scales were referenced to $NO_2CH_3$.

**Powder X-ray diffraction.** X-ray powder patterns were recorded on a STOE Stadi-P diffractometer equipped with a Cu X-ray tube, a Ge(111) monochromator and a Mythen detector. For PN-NAC and fMLF samples, Rietveld refinements were performed with TOPAS Academic-64 V6[84]. The background was treated with a Chebyshev polynomial with 20 parameters.

Full experimental and computational details are provided in Supplementary Methods.

## Data availability

All data generated or analyzed during this study are included in this published article and its supplementary information files. The Supplementary Information includes materials and methods details, DART-HRMS spectra, solution and solid-state NMR data, PXRD patterns and Rietveld refinements, database-mining outputs, GIPAW-calculated NMR parameters, and full comparison tables between experimental and computed chemical shifts for both PN–NAC and fMLF. The MicroED crystal structures reported in this study have been deposited at the Cambridge Crystallographic Data Centre (CCDC), under deposition numbers 2506116 (PN–NAC) and 2506115 (fMLF). These data can be obtained free of charge from the Cambridge Crystallographic Data Centre via www.ccdc.cam.ac.uk/data_request/cif. The DFT-D-optimized structures are provided as separate supplementary files: Supplementary Data 1 and 2 for PN–NAC (optimized with variable and fixed lattice parameters, respectively); and Supplementary Data 3 for fMLF (optimized with variable lattice parameters).

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

## Acknowledgements

M.R.C. and C.S. acknowledge support from the project CH4.0 under the MUR program "Dipartimenti di Eccellenza 2023-2027" (CUP: D13C22003520001), the project FLIPPER (PRIN2022 n. 202224KAX8; CUP D53D23010020006) funded by European Union - Next Generation EU, Mission 4 Component 1, the project NICE (PRIN2020 n. 2020Y2CZJ2; CUP D13C22000440001). F.B. acknowledges funding by the Alexander von Humboldt Foundation.

## Author contributions

Y.N. and M.R.C. conceived the study. Y.N., M.R.C. and C.S. designed the work. C.S. performed solution/solid-state NMR. N.M., M.N. and Y.A. collected MicroED data. K.A., K.K. and M.H. collected high-resolution MS data. F.B. and Y.N. performed DFT-D/GIPAW calculations. F.B. performed PXRD analyses. C.S. and Y.N. performed database analysis. All authors contributed to interpretation of data. C.S., Y.N., M.R.C. wrote the first draft of the manuscript. All authors contributed to discussions and to writing and revising the final draft.

## Competing interests

The authors C.S., F.B., and M.R.C. declare no competing interests. Authors N.M., M.N., Y.A., K.A., K.K., M.H. and Y.N. are full-time employees of JEOL Ltd. The company had no role in the study design, data collection and analysis, decision to publish, or preparation of the manuscript.
