## [Transparent Peer Review file · Communications Chemistry]

An integrated workflow for the structure elucidation of nanocrystalline powders

Corresponding Author: Dr Yusuke Nishiyama

Parts of this file have been redacted as indicated.

Version 0:

Reviewer comments:

Reviewer #1

(Remarks to the Author)

This is a strong paper which provides excellent evidence for the success of a combined microED, mass spec, NMR, and DFT approach to determining the crystal structure of a cocrystal prepared mechanochemically. This combination appears to be novel and the work addresses an important topic, namely how to structurally characterize materials which do not yield crystals suitable for traditional diffraction experiments. The particular example discussed here is obtained mechanochemically but the method is not limited to mechanochemical reaction products. The paper should have good impact in the fields of crystallography and mechanochemistry. I have only a few comments for the authors to consider.

1. The title is a bit awkward, i.e., "mechanochemical nanopowders" doesn't quite make sense. I am also not sure all methods used need to be listed in the title.
2. The title also uses the word "scalable" and this is also mentioned in the conclusions of the paper. I did not find any real motivation for the use of this word. I did not see a discussion or demonstration that the method is 'scalable'; nor what is exactly meant by this.
3. The authors mention in the introduction that "countless attempts" to get a single crystal of the compound had failed. Is this the present work? Past work? Can details be provided on what was tried?
4. Although the current work is very thorough and convincing, it really would have been good to see the method validated on more than a single compound. I realize this will be a bit of work to do. Adding another compound or two is not necessary for publication, but as it stands it is hard to be sure that the method is completely generalizable.

Reviewer #2

(Remarks to the Author)

In this manuscript, Sabena et al. reported a workflow for structure determination of mechanochemically synthesized two-component systems using a combination of. MicroED with high-resolution mass spectrometry (HRMS), database mining, solution and solid-state NMR, and computational methods.

Materials and methods: To demonstrate the feasibility of the workflow the authors used pyridoxine and N-acetyl-L-cysteine (PN-NAC) salt. The authors claim that due to its sticky nature, the crystallization of this salt is difficult. According to the authors, MicroED offers the lattice and skeleton structure of the adduct without clear hydrogen or heteroatom assignments. Further, using refinement based on the molecular mass using mass spectrometry, chemical shift values using solution and solid state NMR spectroscopy and computational studies allowed final structure of the structure.

Relevance and Novelty: Determination of solid state structures are of utmost importance in pharmaceutical science, materials chemistry and structural chemistry. Specifically, when single crystal structure is not available, several approaches including, PXRD, SAXS, SANS, SSNMR, MicroED have been reported. Therefore, the present work provides a highly relevant workflow for structure determination of non-crystalline solids.

The approach presented here using a modular approach or "blind test" is original and applicable to other systems.

Limitation: The authors have now demonstrated using one complex, therefore, it is challenging to know the generic nature of this approach. Some comments in this aspect will be useful. Some focus on the literature is also useful.

Overall, the manuscript is logically structured and clearly presented. However, the manuscript requires a minor revision.

Comments:

1. The authors used PubChem as the main database. However, it is useful to comment on the limitations and other alternative databases such as Reaxys or CAS SciFinder which provides better accuracy?
2. The authors state that " It is worth noting that absolute structure s cannot be determined by the current MicroED analysis as we used kinetic analysis. However, we cannot exclude the possibility of N acetyl D cysteine." It is beneficial to the readers, if the authors can indicate other possible analytical tools that can be used to resolve such ambiguities using both L and D isomers.
3. The authors indeed utilized multiple analytical tools. SSNMR has been well documented in the literature to study cocrystals, polymorphs, polymers, soft materials and gels. It is useful if the authors can discuss also in this context, where crystallization is challenging. E.g.; (1) J. Am. Chem. Soc., 2006, 128 , 9555 —9560; (2) R. K. Harris Solid State Sci., 2004, 6 , 1025 —1037; (3) MR Crystallography , R. K. Harris, R. D. Wasylishen and M. J. Duer, John Wiley & Sons Ltd, Chichester, 2009; (4) Analyst, 2006,131 , 351-373; (5) Soft Matter, 2010, 6, 1748-1757;(6) Soft Matter, 2016,12, 6015-6026; (7) Cryst. Growth Des. 2009, 9, 11, 4710–4719.
4. I suggest the authors to move Figure S15 to the main text and the table to the supporting information.
5. Figure S17 shows the powder XRD of the complex. It is useful to include the XRD patterns of the individual components also.
6. The XRD peaks are really prominent, therefore, it is useful to know whether structure can be refined using Rietveld refinement to further confirm the MircoED structure?. See e.g.: Soft Matter, 2010, 6, 3789-3796. It will improve the workflow and quality of the final structures.
- 7.

Reviewer #3

(Remarks to the Author)

The manuscript presents a workflow for structural analysis based on microcrystal electron diffraction. The intrinsic ambiguities of the method (like low proton signal) are resolved by complementary established techniques (MS, NMR, DFT etc.). To make it short, I think the presentation is nice, the discussion is clear and convincing and an instructive example was chosen. Therefore I recommend publication of the manuscript.

Version 1:

Reviewer comments:

Reviewer #1

(Remarks to the Author)

The authors have gone above and beyond the minimum needed to improve this manuscript. They have added an entirely new example, determining the structure of the MLF peptide. This is very impressive and firmly establishes the generality of the protocol. This is exciting work which should be published as-is.

Reviewer #2

(Remarks to the Author)

In the revised version of the manuscript, the authors have addressed all the reviewers' comments. In the revised version of the manuscript, the authors have provided additional supporting information, including solution- and solid-state NMR, DART-MS, PXRD, and refinement, and have extensively revised the results and discussion. Overall, the revised version has improved significantly. The manuscript is acceptable for publication in Communications Chemistry.

Reviewer #3

(Remarks to the Author)

The authors added significant experimental data to support the generality of the approach. Publication of the manuscript is recommended.

[information redacted]

Please find enclosed the revised version of the manuscript **COMMSCHEM-25-0732-T** entitled “**A workflow for the structural elucidation of nanocrystalline powders: Micro Electron Diffraction, Mass Spectrometry, solution/solid-state NMR, and calculations integration**”, by C. Sabena, F. Bravetti, N. Miyauchi, M. Nakafukasako, Y. Aoyama, K. Asakura, K. Konuma, M. Hashimoto, Y. Nishiyama, and M.R. Chierotti, submitted to *Communications Chemistry*. We have carefully considered and answered all the comments provided by the three reviewers, and the Manuscript and the Supplementary Information were modified accordingly. We have prepared and uploaded the revised highlighted version (*Revised Manuscript_highlighted*), in which changes have been tracked, the non-highlighted version (*Revised Manuscript*) of the manuscript, and the *Revised Supplementary Information* file.

Herein a point-by-point reply to the comments of the reviewers is reported: the reviewers' comments are reported in black, our answers are reported in light blue, and the changes in the main text of the Manuscript and in the Supplementary Information are highlighted in yellow.

Reviewer #1 (Remarks to the Author):

This is a strong paper which provides excellent evidence for the success of a combined microED, mass spec, NMR, and DFT approach to determining the crystal structure of a cocrystal prepared mechanochemically. This combination appears to be novel and the work addresses an important topic, namely how to structurally characterize materials which do not yield crystals suitable for traditional diffraction experiments. The particular example discussed here is obtained mechanochemically but the method is not limited to mechanochemical reaction products. The paper should have good impact in the fields of crystallography and mechanochemistry. I have only a few comments for the authors to consider.

General response: We sincerely thank the reviewer for the valuable comments and constructive suggestions, which have significantly contributed to enhancement the quality and clarity of our manuscript.

1. *The title is a bit awkward, i.e., "mechanochemical nanopowders" doesn't quite make sense. I am also not sure all methods used need to be listed in the title.*

Answer: We thank the reviewer for this valuable suggestion. We agree that “mechanochemical nanopowders” in the original title is not the most precise expression. Accordingly, we have revised the title to improve clarity as follows:

“A workflow for the structural elucidation of nanocrystalline powders: Micro Electron Diffraction, Mass Spectrometry, solution/solid-state NMR, and calculations integration”

2. *The title also uses the word "scalable" and this is also mentioned in the conclusions of the paper. I did not find any real motivation for the use of this word. I did not see a discussion or demonstration that the method is 'scalable'; nor what is exactly meant by this.*

Answer: We appreciate the reviewer's observation. The term “scalable” was originally intended to emphasize that each module of the workflow (e.g., HRMS, database mining, MicroED refinement, SSNMR validation) can be expanded to systems of higher complexity. However, since this concept is not quantitatively demonstrated, we have removed the word “scalable” from both the title (as shown in Question 1) and the conclusions to avoid ambiguity.

3. The authors mention in the introduction that "countless attempts" to get a single crystal of the compound had failed. Is this the present work? Past work? Can details be provided on what was tried?

Answer: We thank the reviewer for this comment. The sentence refers to our own attempts performed during this study and the previous one in which this adduct was firstly obtained (Cossard, A. *et al.* Advanced feature analysis for enhancing cocrystal prediction. *Chemometrics and Intelligent Laboratory Systems* **257**, 105318 (2025)). Specifically, single crystals were attempted via slow solvent evaporation and recrystallization from methanol, ethanol, water, acetone, and mixed solvents. In all cases, the product remained sticky even after complete solvent evaporation, thus yielding products unsuitable for SCXRD. We have clarified this point by adding experimental details about the unsuccessful crystallization trials in the **Supplementary Information (Materials and Methods Section)** as follows:

"Several attempts were made to obtain single crystals suitable for SCXRD through slow solvent evaporation and recrystallization from methanol, ethanol, water, acetone, and mixed solvents. In all cases, the product remained sticky even after completing solvent evaporation."

4. Although the current work is very thorough and convincing, it really would have been good to see the method validated on more than a single compound. I realize this will be a bit of work to do. Adding another compound or two is not necessary for publication, but as it stands it is hard to be sure that the method is completely generalizable.

Answer: We thank the reviewer for this valuable suggestion. We fully agree that validating the workflow on additional compounds would further strengthen its general applicability. Thus, in the revised version, we added a second example, *i.e.*, N-formyl-methionyl-leucyl-phenylalanine (fMLF), to further illustrate the applicability of the workflow. fMLF was selected as a second test compound because, despite its extensive use as a model sample, no crystal structure was available in the literature. The same MicroED-HRMS-database mining-DFT/GIPAW calculations-SSNMR procedure applied to PN-NAC successfully yielded the complete crystal structure of fMLF, confirming the robustness and generalizability of the method to chemically and structurally different systems. Thus, to expand the examples of the approach applicability, we added the following paragraphs in the manuscript and Supplementary Information:

In the **Introduction**:

Figure 1. Chemical structures of PN (left), NAC (middle), fMLF (right), with atom numeration.

"Moreover, to further illustrate that the method is generalizable, the same procedure was applied for the structure elucidation of N-formyl-methionyl-leucyl-phenylalanine (fMLF, Figure 1), a bacterial chemoattractant peptide,^{65,66} whose structure has not yet been reported in literature, despite its widespread use as a model compound in solid-state NMR methodology owing to its well resolved ¹³C and ¹H peaks."

In the Results and Discussions Section:

“To further demonstrate the applicability of the approach, fMLF was taken as a second ‘blind test’. The same procedure applied to PN–NAC was followed. The initial structural information on the sample, treated as an unknown compound (“compound 2”), was obtained from MicroED analysis. This provided a preliminary framework of its structure (Figure S20a in the SI), in which hydrogen atoms were not observed, non-hydrogen atoms were initially represented in gray, and a heavier atom, consistent with sulfur, was identified. The application of DART-HRMS (Figure S21 in the SI) enabled the determination of the molecular formula ($C_{21}H_{31}N_3O_5S$, 437.555 Da), through the observation of the monoisotopic peak at 438.2062 Da, in full agreement with the number of non-hydrogen atoms and the presence of a sulfur atom observed in the MicroED data (Figure S20a in the SI). At this point, database mining was used to assess the identity of compound 2. Since the initial search, based solely on molecular formula, returned 1399 candidate molecules, further filters were applied: in particular, the number of non-hydrogen atoms (30) and relevant substructural features (*i.e.*, the sulfur atom was assumed to be part of either a C-S-C, O-S-C, N-S-C, O-S-N, N-S-N, or O-S-O bond, as the MicroED structure revealed the presence of X-S-X ($X = C, N, O$)). This reduced the number of candidates to 75: 14 for C-S-C, 10 for O-S-C, 42 for N-S-C, 9 for O-S-N, and 0 for N-S-N and O-S-O. The corresponding queries are listed in the SI (Paragraph 3.3; a ± 0.05 Da margin was included) and all the possible isomers are reported in Figure S22 in the SI. A visual comparison between the MicroED molecular skeleton and the remaining candidates allowed to exclude most of them, resulting in three possible candidates: N-formyl-DL-methionyl-DL-leucyl-DL-phenylalanine, N-Formyl-L-methionyl-L-leucyl-L-phenylalanine, and N-formyl-D-methionyl-L-leucyl-L-phenylalanine, which are stereoisomers, differing only in the absolute structure (Figure S23 in the SI). Both N-formyl-DL-methionyl-DL-leucyl-DL-phenylalanine and N-formyl-D-methionyl-L-leucyl-L-phenylalanine can be excluded, because neither a racemic mixture nor a stereochemical arrangement involving different configurations at the three chiral centers is compatible with the relative molecular geometry observed in the MicroED skeleton. If such species were present, multiple chiral environments or configurational disorder would be evident in the structure, which is not observed. As in the previous PN–NAC system, MicroED data were obtained by using kinetic analysis, which does not allow determination of the absolute configuration and therefore does not allow exclusion of the D-D-D stereoisomer. However, the L-L-L stereoisomer is supported by its biological compatibility, leading to the identification of compound 2 as N-Formyl-L-methionyl-L-leucyl-L-phenylalanine. More advanced MicroED analyses, such as dynamic scattering analysis together with fine step data collection, could provide additional information on the absolute structure.⁷⁰ The MicroED-derived structure, with atom assignments refined by the combination of DART-HRMS and the database mining, was then geometry-optimized by using DFT-D calculations with variable lattice parameters. The good agreement between the optimized structure, and unit cell parameters, with the experimental ones (Figure 9 and Table S8 in the SI) and the calculated low RMSDC value (0.198) indicate that the experimental structure is correct. GIPAW calculations were subsequently performed on the DFT-D-optimized structure. Computed 1H , ^{13}C , and ^{15}N SSNMR chemical shifts were compared with experimental ones (assigned through several SSNMR experiments including 1H MAS echo, $2D\ ^1H\{^{13}C\}$ short- and long-range DCP, 1H DQ/ 1H SQ, and $^1H\{^{15}N\}$ DCP; Figures S24–S27 in the SI), yielding 1H , ^{13}C , and ^{15}N RMSE values of 0.6, 3.0 and 3.6 (Figures S28–S30 in the SI), respectively. The slightly higher 1H and ^{13}C RMSE values can be partly attributed to the dynamic behavior of the phenylalanine aromatic ring in fMLF which cannot be considered in the calculations performed at 0 K. Under fast MAS, the phenyl ring undergoes rapid motions, which interfere with the 1H - ^{13}C decoupling and therefore reduce the CP efficiency for aromatic CH carbons.^{75,76} As a result, the signals in the 125–135 ppm region are markedly attenuated or even unobservable, making their experimental assignment, and related 1H assignment by $^1H\{^{13}C\}$ DCP, less robust and leading to a slightly increased RMSE. Importantly, the combination of SSNMR and MicroED data enables to identify this motion as a 180° ring flip. In fact, the phenyl ring is clearly resolved in the ED potential map and would otherwise appear obscured if a continuous rotational motion were present. This observation is fully consistent with previous 2H NMR studies on phenylalanine ring dynamics,⁷⁵ but here the flip is detected without the need for deuterium labelling, highlighting the strength of combining 3D ED with SSNMR. To further verify the effect, RMSE values were recalculated excluding the atoms from the ring (H5/H9 and H6/H8; C4/C5/C6/C8/C9), yielding 0.5 ppm for 1H and 2.3 ppm for ^{13}C (Table S10, Figures S31 and S32 the SI). These

low RMSE values confirm the reliability of the obtained structure despite the dynamical attenuation of the aromatic signals. To further confirm the goodness of the crystal structure of fMLF, Rietveld refinement was also performed. The resulting Rietveld plot is reported in Figure S33 in the SI.

This second example clearly demonstrates the modularity and generalizability of the approach. The general workflow remains the same across different samples; however, the number and depth of analyses performed within each characterization step can be adjusted according to the complexity of the investigated system, in order to extract all the information required for the complete structure elucidation.”

Figure 9. Overlay of the experimental (standard colours) and DFT-D optimized (black) crystal structures of fMLF, with view direction along the a) a-axis and b) b-axis.

In the Conclusions:

“Furthermore, the comprehensive structure elucidation of fMLF, used as a second example of application of the workflow, enabled to evaluate the generalizability of the method.”

In the Supplementary Information (Materials and Methods Section):

“**N-formyl-methionyl-leucyl-phenylalanine (fMLF).** fMLF in powder form was purchased from Sigma-Aldrich and used without further purification.”

“**MicroED.** The ED patterns of the PN-NAC and fMLF crystals were measured using an XtaLAB Synergy-ED (Rigaku corporation and JEOL Ltd., Japan) operating at 200 kV with continuous rotation of the sample. To minimize the electron radiation damage, all of the measurements were performed with a low dose rate of $1 \text{ e}^- \text{ nm}^{-2} \text{ s}^{-1}$ to avoid sample degradation. The samples were kept at 297 K during measurements. The diffraction data were recorded using a high-sensitivity pixel array detector (Hypix-ED, Rigaku corporation, Japan). The camera length (606.590 mm) was calibrated using a gold polycrystal specimen as a standard. While the seven data sets, which were measured from seven different crystals, were merged to obtain 3D structure with high completeness for PN-NAC, a set of data from a single crystal was used for fMLF. A rotation series for all set of diffraction patterns contained ~ 160 frames, which were collected using holder rotation steps of $\sim 0.5^\circ$ and covered a range of $\sim 80^\circ$ over 80 seconds. The diffraction patterns were recorded for crystals of micrometer size ($1 \text{ }\mu\text{m}$ to $2 \text{ }\mu\text{m}$). The crystallographic data for the structures reported in this paper were deposited within the Cambridge Crystallographic Data Centre under the CCDC deposition numbers 2506116 (PN-NAC) and CCDC 2506115 (fMLF). Copies of the data can be obtained free of charge from www.ccdc.cam.ac.uk/data_request/cif.”

“**DFT-D and GIPAW Calculations.** The crystal structure of the PN-NAC salt was optimized at DFT-D level with Quantum Espresso (QE, v. 6.4.1)², employing the projector augmented wave (PAW) approach, with the

non-local vdW-df2 method³ and the B86r functional⁴ with the SSSP set of pseudopotentials.⁵ An energy cut-off of 60 Ry was used. Two different calculations were performed: (1) optimization of atomic positions, keeping the lattice parameters fixed; (2) optimization of both atomic positions and lattice parameters. Starting from the optimized structures with fixed lattice parameters, NMR calculations were performed using the Gauge Including Projected Augmented Wave (GIPAW)⁶ and the PBE pseudopotentials from PS Library 1.0.0⁷ with an energy cut-off of 80 Ry, following the methodology previously described.^{8,9} The theoretical absolute isotropic magnetic shielding (σ_{iso}) values obtained from GIPAW were converted into isotropic chemical shifts (δ_{iso}) using the fixed-slope relation $\delta_{\text{iso}} = \sigma_{\text{ref}} - \sigma_{\text{iso}}$, where σ_{ref} was obtained by minimizing the least-squares deviation between experimental and calculated values, *i.e.*, $\sigma_{\text{ref}} = \langle \sigma_{\text{iso}} \rangle + \langle \delta_{\text{exp}} \rangle$ (Figures S8-S11).¹⁰⁻¹² The σ_{ref} values, obtained from a constrained linear regression (slope -1), were: 30.423/300.336 ppm for ¹H, 166.830 for ¹³C, and 219.568 ppm for ¹⁵N.

The crystal structure of fMLF was obtained by geometry optimization at the DFT-D2 level using QE (v7.5), employing the PAW approach with verified pseudopotentials from the official QE website. An energy cutoff of 47 Ry, as recommended for the pseudopotentials, was applied. Both atomic positions and lattice parameters were optimized, with the lattice parameters changing by approximately 1% after optimization. NMR calculations were performed using the GIPAW method with the same cutoff energy of 47 Ry and processed following the same procedure as described for PN-NAC above (Figures S27-S29). The σ_{ref} values, obtained from a constrained linear regression (slope -1), were: 30.356/30.410 ppm for ¹H, 170.882/170.264 for ¹³C, and -314.425 ppm for ¹⁵N.”

“**Solid-State NMR.** [...] For ¹H MAS spectra and 2D ¹H/{¹⁴N} T-HMQC, ¹H/{¹³C} and ¹H/{¹⁵N} double CP (DCP), and ¹H Double Quantum /¹H Single Quantum (¹H DQ/¹H SQ) experiments (performed at JEOL Ltd., Akishima, Tokyo), PN-NAC and fMLF were packed into a 1 mm zirconia rotor, spun at a MAS frequency of 70 kHz, and optimized recycle delays of 26.4 s and 3.3 were used, respectively. [...] The ¹H/{¹³C} and ¹H/{¹⁵N} DCP experiments were acquired using a double CP sequence (¹H→¹³C→¹H). For ¹H/{¹³C} DCP, two datasets were collected to probe both short- and long-range ¹H-¹³C proximities: the short-range dataset used contact times of 1 ms (ct1) and 0.1 ms (ct2) for both PN-NAC and fMLF, while the long-range dataset employed 1 ms (ct1) and 2 ms (ct2) for PN-NAC and 2 ms (ct1) and 2 ms (ct2) for fMLF.

In the Supplementary Information (Results and Discussion Section):

3. Results and Discussion: fMLF

3.1 MicroED

Figure S20. Asymmetric unit of compound 2 (fMLF). (a) Preliminary MicroED-derived structure of compound 2: carbon, nitrogen, and oxygen atoms are undifferentiated (gray), hydrogen atoms are not visible, and a heavier atom consistent with a sulfur atom (yellow) is identified. (b) Refined structure after integration

of HRMS, database filtering, and NMR analysis. The molecular species is assigned to N-Formyl-L-methionyl-L-leucyl-L-phenylalanine, with all non-hydrogen atoms correctly labeled and colored.

3.2 DART-HRMS

Figure S21. DART-HRMS spectra of compound 2. Ionization mode DART⁺.

3.3 Structure Database Analysis

Queries used:

- C-S-C:

https://pubchem.ncbi.nlm.nih.gov/#query=CSC&tab=substructure&input_type=smiles&mw_gte=437.505&mw_lte=437.605&heavycnt_gte=30&heavycnt_lte=30&fullsearch=true&page=1

- O-S-C:

https://pubchem.ncbi.nlm.nih.gov/#query=OSC&tab=substructure&input_type=smiles&mw_gte=437.505&mw_lte=437.605&heavycnt_gte=30&heavycnt_lte=30&fullsearch=true&page=1

- N-S-C:

https://pubchem.ncbi.nlm.nih.gov/#query=NSC&tab=substructure&input_type=smiles&mw_gte=437.505&mw_lte=437.605&heavycnt_gte=30&heavycnt_lte=30&fullsearch=true&page=1

- O-S-N:

https://pubchem.ncbi.nlm.nih.gov/#query=OSN&tab=substructure&input_type=smiles&mw_gte=437.505&mw_lte=437.605&heavycnt_gte=30&heavycnt_lte=30&fullsearch=true&page=1

- N-S-N:

https://pubchem.ncbi.nlm.nih.gov/#query=NSN&tab=substructure&input_type=smiles&mw_gte=437.505&mw_lte=437.605&heavycnt_gte=30&heavycnt_lte=30&fullsearch=true&page=1

- O-S-O:

https://pubchem.ncbi.nlm.nih.gov/#query=OSO&tab=substructure&input_type=smiles&mw_gte=437.505&mw_lte=437.605&heavycnt_gte=30&heavycnt_lte=30&fullsearch=true&page=1

(C-S-C)

(O-S-C)

(N-S-C)

(O-S-N)

Figure S22. Possible candidates for compound **2** after the application of filters (molecular weight, number of non-hydrogen atoms and relevant substructural features).

Figure S23. Possible candidates for compound **2** after the application of filters and MicroED skeleton visualization.

3.4 SSNMR

Figure S24. Comparison of (a) ^1H (600.1 MHz) MAS echo SSNMR spectrum of fMLF, acquired at room temperature at a spinning speed of 70 kHz, and ^1H projections of (b) 2D ^1H DQ/ ^1H SQ, (c) ^1H - ^{15}N DCP, (d) ^1H - ^{13}C DCP short range, (e) ^1H - ^{13}C DCP long range (atom numeration refers to Figure 1).

Figure S25. (a) 2D ^1H - ^{13}C short-range DCP and (b) 2D ^1H - ^{13}C long-range DCP SSNMR spectra of fMLF, acquired at room temperature at a spinning speed of 70 kHz. Atom numeration refers to Figure 1.

Figure S26. 2D ^1H DQ/ ^1H SQ MAS SSNMR spectrum of fMLF, acquired at room temperature at a spinning speed of 70 kHz. Atom numeration refers to Figure 1.

Figure S27. ^1H - ^{15}N (60.81 MHz) DCP spectrum of fMLF, acquired at room temperature at a spinning speed of 70 kHz. The ^{15}N and ^{14}N chemical shifts are referenced to NO_2CH_3 and atom numeration refers to Figure 1.

3.5 DFT-D and GIPAW Calculations

Table S8. Comparison between experimental and optimized lattice parameters for the crystal structure of fMLF. The calculated RMSDC is also reported.

	Experimental structure	DFT-D optimized
Temperature / K	297	0
Space group	$P 2_1 2_1 2_1$	$P 2_1 2_1 2_1$
a / Å	5.4124(9)	5.4678
b / Å	20.836(5)	20.9155
c / Å	22.219(11)	22.5801
α / °	90	90
β / °	90	90
γ / °	90	90
Volume / Å ³	2505.7	2582.5
RMSDC	0.198	

Table S9. Experimental (exp) and computed (calc) ^1H , ^{13}C and ^{15}N SSNMR chemical shifts (ppm) for fMLF, with assignments (referred to Figure 1). The ^{15}N chemical shifts are referenced to NO_2CH_3 .

^1H SSNMR			
Atom		fMLF exp	fMLF calc
29		14.3	16.0
25		9.2	10.0
21		8.9	9.4
23		7.8	8.4
27		6.9	7.1
6/8		6.7	7.9/5.4
5/9		6.6	6.9/6.5
7		6.3	6.4
17		6.0	6.4
2		5.1	5.4
11		4.4	4.3
19		2.5	2.7
20		2.1	2.5/1.8/1.6
3		2.1	2.1/1.6
18		2.1	2.1/1.7
12		2.1/1.2	2.0/0.9
13		1.9	1.6
14/15		1.6/1/0.9/0.8	1.6/0.9/0.8/0.7/0.1/-0.4
^{13}C SSNMR			
Group	Atom	fMLF exp	fMLF calc
C = O	10	174.6	177.1
COOH	1	173.0	179.5
C = O	16	171.7	173.4
C = O	21	164.9	166.4
C _{q-ar} /CH _{ar}	4/9/5/8/6	135.4	140.3/133.8/131.4/131.2/130.4
CH _{ar}	7	127.4	130.0
CH	11	56.4	57.3
CH	2	54.0	55.7
CH	17	51.7	52.4
CH ₂	12	40.4	40.1
CH ₂	18	37.3	38.1
CH ₂	3	36.5	36.7
CH ₂	19	28.3	29.0
CH	13	24.7	24.7
CH ₃	14/15	24.3	21.0
CH ₃	14/15	19.2	14.2
CH ₃	20	13.7	12.4
^{15}N SSNMR			
Group	Atom	fMLF exp	fMLF calc
NH	23	-250.6	-255.0
NH	25	-260.1	-260.3
NH	27	-268.6	-264.1

Figure S28. Correlation between computed and experimental ^1H chemical shifts for fMLF. The shieldings were converted to chemical shifts using a reference value of 30.356 ppm obtained from a constrained linear regression (slope -1).

Figure S29. Correlation between computed and experimental ^{13}C chemical shifts for fMLF. The shieldings were converted to chemical shifts using a reference value of 170.882 ppm obtained from a constrained linear regression (slope -1).

Figure S30. Correlation between computed and experimental ^{15}N chemical shifts for fMLF. The shieldings were converted to chemical shifts using a reference value of -314.425 ppm obtained from a constrained linear regression (slope -1).

Table S10. Comparison of experimental (exp) and calculated (calc) ^1H chemical shifts for fMLF obtained removing H5/H9 and H6/H8, C4/C5/C6/C8/C9 (atoms from the phenylalanine ring). Assignments refer to Figure 1.

^1H SSNMR			
Atom	fMLF exp	fMLF calc	
29	14.3	16.0	
25	9.2	10.0	
21	8.9	9.4	
23	7.8	8.4	
27	6.9	7.1	
7	6.3	6.4	
17	6.0	6.4	
2	5.1	5.4	
11	4.4	4.3	
19	2.5	2.7	
20	2.1	2.5/1.8/1.6	
3	2.1	2.1/1.6	
18	2.1	2.1/1.7	
12	2.1/1.2	2.0/0.9	
13	1.9	1.6	
14/15	1.6/1/0.9/0.8	1.6/0.9/0.8/0.7/0.1/-0.4	
^{13}C SSNMR			
Group	Atom	fMLF exp	fMLF calc
C = O	10	174.6	175.7
COOH	1	173.0	178.1
C = O	16	171.7	172.0
C = O	21	164.9	165.0
CH _{ar}	7	127.4	128.8
CH	11	56.4	56.5
CH	2	54.0	54.8
CH	17	51.7	51.5
CH ₂	12	40.4	39.3
CH ₂	18	37.3	37.3
CH ₂	3	36.5	35.9
CH ₂	19	28.3	28.2
CH	13	24.7	24.0
CH ₃	14/15	24.3	20.3
CH ₃	14/15	19.2	13.5
CH ₃	20	13.7	11.7

Figure S31. Correlation between computed and experimental ^1H chemical shifts for fMLF obtained removing H5/H9 and H6/H8 (hydrogen atoms from the phenylalanine ring). The shieldings were converted to chemical shifts using a reference value of 30.410 ppm obtained from a constrained linear regression (slope -1).

Figure S32. Correlation between computed and experimental ^{13}C chemical shifts for fMLF obtained removing C4/C5/C6/C8/C9 (carbon atoms from the phenylalanine ring). The shieldings were converted to chemical shifts using a reference value of 170.264 ppm obtained from a constrained linear regression (slope -1).

3.6 PXRD

Figure S33. Rietveld plot of fMLF crystal structure. Black dots: experimental pattern; red dots: calculated fit; gray line: difference curve. Possible peak positions are marked with vertical blue ticks.

Reviewer #2 (Remarks to the Author):

In this manuscript, Sabena et al. reported a workflow for structure determination of mechanochemically synthesized two-component systems using a combination of. MicroED with high-resolution mass spectrometry (HRMS), database mining, solution and solid-state NMR, and computational methods.

Materials and methods: To demonstrated the feasibility of the workflow the authors used pyridoxine and N-acetyl-L-cysteine (PN–NAC) salt. The authors claim that due to its sticky nature, the crystallization of this salt is difficult.

According to the authors, MicroED offers the lattice and skeleton structure of the adduct without clear hydrogen or heteroatom assignments. Further, using refinement based on the molecular mass using mass spectrometry, chemical shift values using solution and solid state NMR spectroscopy and computational studies allowed final structure of the structure.

Relevance and Novelty: Determination of solid state structures are of utmost importance in pharmaceutical science, materials chemistry and structural chemistry. Specifically, when single crystal structure is not available, several approaches including, PXRD, SAXS, SANS, SSNMR, MicroED have been reported. Therefore, the present work provides a highly relevant workflow for structure determination of non-crystalline solids.

The approach presented here using a modular approach or "blind test" is original and applicable to other systems.

General response: We sincerely thank the reviewer for the constructive feedback and comments, which have helped us to improve the overall quality of our manuscript.

Limitation: The authors have now demonstrated using one complex, therefore, it is challenging to know the generic nature of this approach. Some comments in this aspect will be useful. Some focus on the literature is also useful.

Answer: We thank the reviewer for raising this important point concerning the general applicability of the workflow. As detailed in our response to Reviewer #1 (Comment 4), we have now included a second model system, *i.e.*, N-formyl-methionyl-leucyl-phenylalanine (fMLF), to further demonstrate the generalizability of the method. This additional example highlights that the proposed workflow can be successfully applied from single to multicomponent systems, confirming its broader potential for structural characterization of microcrystalline and complex molecular solids.

Overall, the manuscript is logically structured and clearly presented. However, the manuscript requires a minor revision.

Comments:

1. The authors used PubChem as the main database. However, it is useful to comment on the limitations and other alternative databases such as Reaxys or CAS SciFinder which provides better accuracy?

Answer: We thank the reviewer for this useful suggestion. We agree that PubChem has limitations in terms of molecular weight precision and query flexibility. In particular, PubChem does not allow simultaneous constraints on molecular formula and substructures, which may result in redundant or imprecise hits. Nonetheless, it is important to note that PubChem offers the advantage of being an open and freely accessible database, making it suitable for workflows intended to be broadly applicable (from academic to industry use), whereas other databases such as Reaxys or CAS SciFinder, although more accurate and versatile, require paid access. However, to clarify the limitations of using PubChem, we have revised the following sentence in the **Results and Discussion** Section:

“This inconsistency arises from the limited precision of the molecular-weight data and the query constraints available in PubChem, *e.g.*, it is not possible to simultaneous constraints on molecular formula and substructures. Databases offering higher numerical resolution and the possibility to combine molecular formula and structural substructure filters (such as Reaxys or CAS SciFinder) could overcome this limitation and yield more accurate and specific results, although access requires a paid license.”

Furthermore, an intrinsic and general limitation of database-based approaches is that a molecular species may not be present in the database at all, especially in the case of newly synthesized or previously unreported compounds. To address these issues, we added a discussion regarding how machine-learning algorithms could be used to generate and rank candidate molecular structures consistent with experimental constraints obtained from MicroED, HRMS, and NMR data, prior to comparison with database entries. This integration would further expand the applicability of the workflow to systems containing previously unreported molecules. Thus, we added in the **Conclusions** the following sentences:

“Additionally, an intrinsic limitation of database-based approaches is that previously unreported molecules may simply be absent from the database, leading to incomplete or inconclusive searches. In such cases, the workflow could be extended with machine-learning-assisted structure generation, where algorithms propose and rank molecular candidates that satisfy the experimental constraints derived from MicroED, HRMS, and NMR analyses. These predicted structures could then be cross-validated against chemical structure databases such as PubChem, Reaxys, ChemSpider, or CAS SciFinder, thereby broadening the applicability of the method to unknown or novel compounds.”

2. *The authors state that " It is worth noting that absolute structure cannot be determined by the current MicroED analysis as we used kinetic analysis. However, we cannot exclude the possibility of N acetyl D cysteine." It is beneficial to the readers, if the authors can indicate other possible analytical tools that can be used to resolve such ambiguities using both L and D isomers.*

Answer: We thank the reviewer for this insightful comment. As correctly noted, the present MicroED data were obtained by using kinetic analysis, which does not allow direct determination of the absolute configuration. To resolve such ambiguities without introducing additional analytical techniques, we suggest performing further MicroED measurements based on dynamical scattering refinement combined with fine-step data collection. This is the approach we recommend, as it can provide sensitivity to the absolute structure and enable the distinction between enantiomeric forms within the same experimental framework. However, to better clarify this point we have revised the text in the **Results and Discussion** Section as follows:

“Further analyses, such as dynamic scattering analysis together with fine step MicroED data collection, would be required to address this issue, as these approaches can directly provide information on the absolute configuration without the need for additional analytical methods.”⁶⁸

3. *The authors indeed utilized multiple analytical tools. SSNMR has been well documented in the literature to study cocrystals, polymorphs, polymers, soft materials and gels. It is useful if the authors can discuss also in this context, where crystallization is challenging. E.g.; (1) J. Am. Chem. Soc., 2006, 128, 9555—9560; (2) R. K. Harris Solid State Sci., 2004, 6, 1025—1037; (3) MR Crystallography, R. K. Harris, R. D. Wasylishen and M. J. Duer, John Wiley & Sons Ltd, Chichester, 2009; (4) Analyst, 2006, 131, 351-373; (5) Soft Matter, 2010, 6, 1748-1757; (6) Soft Matter, 2016, 12, 6015-6026; (7) Cryst. Growth Des. 2009, 9, 11, 4710–4719.*

Answer: We thank the reviewer for this insightful suggestion, and we agree that this addition strengthens the rationale for using SSNMR as a complementary method when crystallization is inherently difficult. We have expanded the **Introduction** as follows to include a short discussion on the broader applicability of SSNMR to poorly crystalline or amorphous systems such as polymers, gels, and soft materials, supported by the suggested references:

“Beyond crystalline solids, SSNMR has also proven invaluable for characterizing partially ordered or amorphous systems where crystallization is challenging, such as polymers, gels, or soft materials, due to its sensitivity to local structure and hydrogen bonding^{30–36}”

4. I suggest the authors to move Figure S15 to the main text and the table to the supporting information.

Answer: We agree with the reviewer’s suggestion. Figure S15, showing the ^{13}C CPMAS spectra of PN, NAC, and PN–NAC, provides direct experimental confirmation of the 1:1 stoichiometric ratio and the representativeness of the bulk. We have therefore moved it to the main text (now **Figure 6**), and the associated numerical data have been moved in the Supplementary Information as **Table S6**.

5. Figure S17 shows the powder XRD of the complex. It is useful to include the XRD patterns of the individual components also.

Answer: We thank the reviewer for this suggestion. We have now added the PXRD patterns of pure PN and NAC in the Supplementary Information alongside that of PN–NAC (**Figure S17**). This comparison highlights the significant changes in diffraction patterns, confirming the formation of a new crystalline phase distinct from the starting materials. No evidence of the presence of traces of the starting materials was found in the experimental powder pattern PN–NAC, confirming the purity of the obtained adduct.

Figure S17. Overlay of PXRD diffractograms: experimental PN–NAC (black), experimental pure PN (red), and experimental pure NAC (blue).

6. The XRD peaks are really prominent, therefore, it is useful to know whether structure can be refined using Rietveld refinement to further confirm the MicroED structure?. See e.g.: *Soft Matter*, 2010, 6, 3789-3796. It will improve the workflow and quality of the final structures.

Answer: We thank the reviewer for this suggestion. We have now added the Rietveld plot of the PN-NAC structure (**Figure S19 in the SI**). The Rietveld refinement further confirms the goodness of the crystal structure obtained by MicroED.

Figure S19. Rietveld plot of PN-NAC crystal structure. Black dots: experimental pattern; red dots: calculated fit; gray line: difference curve. Possible peak positions are marked with vertical blue ticks.

The following sentences were added in the main text (**Results and Discussion** Section):

“To further confirm the crystal structure of PN-NAC obtained by MicroED, a Rietveld refinement was performed. All details about refined parameters are given in the Materials and Methods section in the SI, while the Rietveld plot is reported in Figure S19 in the SI. The Rietveld refinement confirms that the determined crystal structure for PN-NAC is correct.”

The following paragraph was added to the **Materials and Methods** section in the **Supplementary Information**:

“X-ray powder patterns were recorded on a STOE Stadi-P diffractometer equipped with a Cu X-ray tube, a Ge(111) monochromator and a Mythen detector. The powders were filled into glass capillaries with 1.0 mm inner diameter. The capillary was spun during the measurement. Cu-K α_1 radiation was used, covering a 2 θ range of 3–80°. The PSD step size was 0.5° with a measurement time of 60 s per step.

For PN-NAC and fMFL samples, Rietveld refinements were performed with TOPAS Academic-64 V6.¹³ The background was treated with a Chebyshev polynomial with 20 parameters. The peak profile was described by the fundamental parameter approach refining crystal size and strain, too. One overall isotropic displacement parameter (B_{iso}) was used for all atoms except hydrogens, for which B_{iso} was assumed to be higher by a factor of 1.2. a correction for were not necessary. Spherical harmonics of the 6th order were used, to correct preferred orientation and anisotropic peak broadening.”

Reviewer #3 (Remarks to the Author):

The manuscript presents a workflow for structural analysis based on microcrystal electron diffraction. The intrinsic ambiguities of the method (like low proton signal) are resolved by complementary established techniques (MS, NMR, DFT etc.). To make it short, I think the presentation is nice, the discussion is clear and convincing and an instructive example was chosen. Therefore I recommend publication of the manuscript.

Answer: We sincerely thank the reviewer for their positive assessment and encouraging comments. We are pleased that the overall presentation and discussion were found clear and convincing, and that the chosen example was considered appropriate to illustrate the workflow. We appreciate the reviewer's recommendation for publication.

Reviewer #1 (Remarks to the Author):

The authors have gone above and beyond the minimum needed to improve this manuscript. They have added an entirely new example, determining the structure of the MLF peptide. This is very impressive and firmly establishes the generality of the protocol. This is exciting work which should be published as-is.

General response: We sincerely thank the reviewer for their positive assessment and encouraging comments.

Reviewer #2 (Remarks to the Author):

In the revised version of the manuscript, the authors have addressed all the reviewers' comments. In the revised version of the manuscript, the authors have provided additional supporting information, including solution- and solid-state NMR, DART-MS, PXRD, and refinement, and have extensively revised the results and discussion. Overall, the revised version has improved significantly. The manuscript is acceptable for publication in Communications Chemistry.

General response: We sincerely thank the reviewer for their positive assessment and encouraging comments.

Reviewer #3 (Remarks to the Author):

The authors added significant experimental data to support the generality of the approach. Publication of the manuscript is recommended.

General response: We sincerely thank the reviewer for their positive assessment and encouraging comments.